# Microscopic Analysis of Heterochromatin, Euchromatin and Cohesin in Cancer Cell Models and under Anti-Cancer Treatment

**Elias Ferdinand Fischer, Götz Pilarczyk and Michael Hausmann ***

Kirchhoff-Institute for Physics, Heidelberg University, Im Neuenheimer Feld 227, 69120 Heidelberg, Germany; fischer.elias@arcor.de (E.F.F.); goetz.pilarczyk@kip.uni-heidelberg.de (G.P.)
*   Correspondence: hausmann@kip.uni-heidelberg.de

**Abstract:** The spatial organization of euchromatin (EC) and heterochromatin (HC) appears as a cell-type specific network, which seems to have an impact on gene regulation and cell fate. The spatial organization of cohesin should thus also be characteristic for a cell type since it is involved in a TAD (topologically associating domain) formation, and thus in gene regulation or DNA repair processes. Based on the previous hypotheses and results on the general importance of heterochromatin organization on genome functions in particular, the configurations of these organizational units (EC represented by H3K4me3-positive regions, HC represented by H3K9me3-positive regions, cohesins) are investigated in the cell nuclei of different cancer and non-cancerous cell types and under different anti-cancer treatments. Confocal microscopic images of the model cell systems were used and analyzed using analytical processes of quantification created in Fiji, an imaging tool box well established in different fields of science. Human fibroblasts, breast cancer and glioblastoma cells as well as murine embryonal terato-carcinoma cells were used as these cell models and compared according to the different parameters of spatial arrangements. In addition, proliferating, quiescent and from the quiescent state reactivated fibroblasts were analyzed. In some selected cases, the cells were treated with X-rays or azacitidine. Heterogeneous results were obtained by the analyses of the configurations of the three different organizational units: granulation and a loss of H3K4me3-positive regions (EC) occurred after irradiation with 4 Gy or azacitidine treatment. While fibroblasts responded to irradiation with an increase in cohesin and granulation, in breast cancer cells, it resulted in decreases in cohesin and changes in granulation. H3K9me3-positive regions (HC) in fibroblasts experienced increased granulation, whereas in breast cancer cells, the amount of such regions increased. After azacitidine treatment, murine stem cells showed losses of cohesin and granulation and an increase in the granulation of H3K9me3-positive regions. Fibroblasts that were irradiated with 2 Gy only showed irregularities in structural amounts and granulation. Quiescent fibroblasts contained less euchromatin-related H3K4me3-positive signals and cohesin levels as well as higher heterochromatin-related H3K9me3-positive signals than non-quiescent ones. In general, fibroblasts responded more intensely to X-ray irradiation than breast cancer cells. The results indicate the usefulness of model cell systems and show that, in general, characteristic differences initially existing in chromatin and cohesin organizations result in specific responses to anti-cancer treatment.

**Keywords:** heterochromatin organization; euchromatin organization; cohesin organization; confocal microscopy; cancer model cell systems; radiation treatment; azacitidine treatment

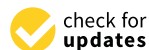



## 1. Introduction

It has become textbook knowledge that the organization of chromatin in cell nuclei is not random but functionally correlated. The dominating view has been that euchromatin and heterochromatin form separated compartments, which are relatively independent from each other [1].

Euchromatin is less condensed, more accessible to proteins, gene rich and easier to transcribe [2]. The regulatory sequences in these regions are accessible to nucleases and are

characterized by the fact that the core histones H3 and H4 are hyper-acetylated, leading to an opening of the chromatin fiber [3]. In addition, a tri-methylation of lysine K4 is present in histone H3 (H3K4me3), which functions as a histone marker [4]. This methylation correlates positively with the transcription of these genes.

Heterochromatin, on the other hand, is typically highly condensed, more inaccessible to proteins and highly organized [2]. A further distinction is made between constitutive and facultative heterochromatin. Those chromosomal regions that have a high density of repetitive DNA elements, such as satellite sequences and transposons, form the constitutive heterochromatin. They remain condensed throughout the cell cycle and should be very similar in different cell types. Facultative heterochromatin contains down-regulated gene loci. The gene loci that belong to facultative heterochromatin can differ in different cell types [5]. In both classes of heterochromatin, histones are hypo-acetylated, resulting in the inaccessibility of DNA to transcription machinery [4]. In addition, constitutive heterochromatin is characterized by the tri-methylation of lysines K9 and K20 in histone H3 (H3K9me3, H3K20me3). Facultative heterochromatin has the tri-methylation of lysine K27 in histone H3 (H3K27me3).

With regard to DNA methylation, cancer development is characterized by, firstly, a general demethylation of DNA in the genome, which leads to significantly increased gene activation and chromosome breaks, and, secondly, the simultaneous increase in DNA methylation at promoters of tumor suppressor genes resulting in their deactivation [6]. These mechanisms lead to increased cell proliferation and uncontrolled tumor growth. The deactivation of tumor suppressor genes offers possibilities for the development of drugs that interfere with the DNA methylation process, such as azacitidine [7].

In addition to these epigenetic modifications, some recent data based on self-organization principles [8,9] point to the leading role of heterochromatin in genome maintenance, highlighting the attractions between heterochromatic regions as being central to the phase separation of the active and inactive chromatin domains [10]. With its main properties, which are silencing effects on transcription around its local positions, its stickiness and its flexible rigidity, heterochromatin forms an important network that impacts cell nuclear functioning by modifying open spaces for molecular diffusion and trafficking to function relevant interaction points [9,11]. The physical features of constitutive heterochromatin, such as, for instance, the regular organization or the condensation, are conformed by a set of proteins, such as cohesin and others. Thereby, it is assumed that cohesin catalyzes the folding of the genome by a process known as "loop extrusion". This involves the formation of chromatin loops by cohesin clasps of two chromatin strands. The resulting loops are then gradually enlarged as cohesin moves along the chromatin strands [12,13]. This process of TAD (topologically associated domain) formation [14] allows different DNA segments that are far apart on the linear strand to be brought into spatial proximity.

The topology of heterochromatin can provide a 3D space for genome activity in addition to the DNA activity coding function. This topological effect may impact whole genome regulation. Recently, it has been shown that, after the stimulation of cells with neuregulin, the timing of structural changing of pericentromere-associated domains (PADs) coincides with the activation of gene expression occurring by critical self-organization in cancer cells and the biphasic activation of stress-response genes of the FOS family [15]. Heterochromatin silencing is changed by splitting PAD clusters and can induce euchromatin re-organization [15].

These studies have shown that genetic activities require the system to act and react as a whole. However, beyond these findings, the questions of how this non-random organization is formed in the closed volume of a cell nucleus, how a tissue specificity is implemented, what the dynamics of chromatin organization means for cell differentiation or cell fate, etc., are still under investigation. Recently, state-of-the-art attempts for an integrative approach to genome expression regulation and response to environmental stress have been reviewed [16]. In cases of environmental stress, or during differentiation

towards specialized cell types or to dysfunctional tumor cells, the chromatin in a cell nucleus seems to react as a whole.

Therefore, a general hypothesis assumes that chromatin is self-organizing according to the physical laws of electrodynamics and thermodynamics under given constraints of the closed spherical volume of the cell nucleus in an environment of water and other molecular compounds. Based on this hypothesis, changes in the molecular compounds in a cell nucleus would induce re-organization of chromatin as a whole. This would change the accessibility of chromatin regions and the spatial neighborhood of given chromatin domains, which should result in different activity responses. In the following sections, we present some of these characteristic responses as being quantified by diffraction-limited fluorescence microscopy. In contrast to the other articles where we analyze chromatin organization on the nanoscale using super-resolution localization microscopy (see, for example, [9]), in this study, we focus on the changes in the microscale, i.e., only prominent changes in chromatin organization that can be quantified by the type of microscopy used. Due to point spreading, the chosen instrument averages over small biological fluctuations that usually overlay the major organization and re-organization processes.

## 2. Materials and Methods

### 2.1. Cell Lines and Cell Preparation

Four different, commercially available cell lines (ATCC, LGC Standards GmbH, Wesel, Germany) were used were used: CRL-2522 (human), P19 (mouse), SkBr3 (human) and U87 MG (human):

CRL-2522 is a human fibroblast cell line derived from male foreskin [17]. Among the cell lines used for this work, it was the only one that was not a cancer cell line. Fibroblasts are mesenchymal connective tissue cells that play an important role in epithelial–mesenchymal interactions. They secrete various growth factors and cytokines that directly affect epidermal proliferation, differentiation and formation of the extracellular matrix.

U87 MG represents a human primary glioblastoma cell line that is frequently used in brain tumor research, since glioblastoma is the most common primary brain tumor. The cells have an adherent epithelial morphology [18].

SkBr3 is a human breast cancer cell line originally derived from adenocarcinoma of the mammary gland tissue. The cells exhibit an epithelial morphology in tissue culture. They overexpress the gene product HER2/neu from the epithelial growth factor receptor family, which is involved in various breast cancer proliferation pathways [19].

P19 is a pluripotent embryonal carcinoma cell line derived from a murine embryonal terato-carcinoma. It possesses a number of stem cell properties, such as the ability to differentiate into cell types from all three germ layers, which makes it a suitable model system. The cells exhibit a normal mouse karyotype suggesting that they do not have gross genetic abnormalities [20,21].

All cells were cultivated according to the protocols specified by the manufacturer ATCC for the respective cell lines [17,22–24]. The cultivation was followed by the preparation and fixation of the cells.

This resulted in six preparations of the CRL-2522 cell line, one of U87 MG and two each of SKBR3 and P19.

The first preparation consisted of formaldehyde (obtained freshly from paraformaldehyde)-fixed proliferating fibroblasts of the CRL-2522 cell line ("CRL2522run"). In addition, the fibroblasts were seeded for 260 d in cell culture flasks. No passage was performed during this entire period. The cell culture medium was renewed every 14 d, and a pH indicator in the medium was used to ensure that neither starving nor acidification occurred during the long-term culture.

A total of 2 further preparations were produced after 260 d in the cell culture flask: for 1 preparation, these resting cells were transferred to a glass slide and fixed. This was the preparation of the quiescent ("resting") cells ("CRL2522dorm"). For the other preparation, the cells were seeded on a glass slide. After about two additional cell cycles,

which quadrupled the confluence, they were also fixed. This preparation thus consisted of reactivated ("awakened") cells ("CRL2522wake"). This long-term culture over 260 d was aimed at creating a tissue model, since in natural tissue, cell division and growth are regulated by signals from the extracellular matrix and contact inhibition, which is why the cells stop proliferating when space is no longer available. This is not the case in conventional cell culture [25].

The fourth and fifth CRL-2522 preparations were each irradiated with X-rays at a dose of 2 Gy. The 4th preparation ("CRL2522_2Gy,1.5h") was fixed 1.5 h post-irradiation and the 5th one ("CRL2522_2Gy,8h") 8 h post-irradiation. The last CRL2522 preparation was irradiated with X-rays at a radiation dose of 4 Gy and fixed 1.5 h post-irradiation ("CRL2522_4Gy,1.5h").

For U87 MG, SKBR3 and P19, an untreated preparation was fixed as for "CRL2522run" ("U87", "SKBR3_0Gy", "P19Aza−"). Furthermore, SKBR3 cells were irradiated with X-rays at a radiation dose of 4 Gy and fixed 1.5 h post-irradiation ("SKBR3_4Gy,1.5h"). For the last preparation, P19 cells were treated with azacitidine at a concentration of 100 nM before fixation ("P19Aza+").

For microscopy, all cell nuclei were stained with DAPI and labeled with different primary and secondary antibodies (ThermoFisher Scientific, Schwerte, Germany), according to the manufacturers' protocols. In order to avoid inter-assay variations, all slides were labeled simultaneously in one common approach. The following combinations were applied: rabbit anti-H3K4me3 (mostly representing euchromatin) with Alexa647-donkey anti-rabbit, mouse anti-H3K9me3 (mostly representing heterochromatin) with Alexa555-goat anti-mouse and goat anti-cohesin with Alexa491-donkey anti-goat. After labeling, the cells were embedded in ProLong Gold antifade embedding solution (ThermoFisher scientific, Schwerte, Germany), sealed and stored at 4 °C until ProLong Gold was polymerized.

## 2.2. Rational for the Experiments

For each dataset, the cell nuclei were selected from low-background regions of 3–5 slides after a quality check of the preparations. By this preselection, representative datasets were obtained and different aspects were considered under which the preparations could be compared. CRL2522run, CRL2522dorm and CRL2522wake could be compared in terms of their cell states. The cells CRL2522run were proliferating, CRL2522dorm were quiescent and CRL2522wake started to proliferate from the quiescent state. CRL2522run, CRL2522_2Gy,1.5h and CRL2522_4Gy,1.5h were compared according to treatment with X-rays of different doses, but fixation after the same repair time. CRL2522_2Gy,1.5h and CRL2522_2Gy,8h were compared with respect to the different repair times at the same radiation dose.

SKBR3_0Gy and SKBR3_4Gy,1.5h were compared in relation to irradiation with X-rays. This was relevant with regard to radiation therapy in oncological situations. Another possibility for a comparison was SKBR3_4Gy,1.5h and CRL2522_4Gy,1.5h. Both preparations were subjected to the same radiation dose. This is interesting as radiation therapy for a tumor also affects the surrounding healthy tissue, such as connective tissue with fibroblasts, which may also be exposed to radiation.

CRL2522run could also be compared with the untreated preparations of the cancer cell lines (U87, SKBR3_0Gy, P19Aza−) as all preparations were not treated; CRL2522run differed from the cancer cells as they were non-transformed cells. A comparison of P19Aza− and P19Aza+ was required because the cells were treated or not treated with azacitidine. The treated cells showed a reduced methylation pattern compared to the untreated cells, and thus should have had an altered gene expression.

## 2.3. Confocal Microscopy and Image Analysis

For data acquisition, a Nikon C2 Plus confocal microscope was used. The images were taken with the Nikon Apo60x Oil λS DIC N2 oil immersion objective and lasers of wavelengths of 405 nm (DNA-DAPI), 488 nm (cohesin), 561 nm (H3K9me3) and 640 (H3K4me3),

all set to 10% of the relative intensity. In addition, the scanner zoom was set to a 4x magnification. The parameters set were a scan size of a 2048 × 2048-pixel resolution with a scan speed of 0.03, resulting in a pixel dwell time of 5.5 µs. The length of an image pixel corresponded to 25.88 nm and thus 1 pixel corresponded to 669.77 nm$^2$. In most cases, between 43 and 80 cell nuclei were recorded. The images obtained in this way were subsequently analyzed for evaluation with Fiji (Version 2.1.0 [26]). After an interactive inspection concerning the completeness and labeling quality as well as exclusion by the analysis software, typically between 15 and 70 cell nuclei were evaluated in each experiment.

In the first step, background signals outside the nuclei were eliminated using the DAPI labeling channel for segmentation. Only tools implemented into Fiji were applied. In brief: for segmentation, the separate images of the color channels were merged. The scaling was quantified by the scale bars that were automatically added to the images by the microscope acquisition software. With the Fiji analysis tool "Area" and "Mean Gray Value", the sizes of nuclei were determined in µm$^2$ and the relative integrated intensity was measured in arbitrary units of 0–4095. In addition, the granularities of the labeling signals of cohesin and H3K9me3 were determined according to number of granules, mean size and mean intensity. These parameters of granularity were calculated in Fiji using several program tools available. After transferring the image scaling into pixel scaling, the intensity histogram of the respective image was smoothed into a bimodal distribution with two local maxima j and k. Then, an intensity threshold t was calculated for each image separately: t = 0.5 (j + k). All pixel intensity values > t were assigned as the pixels of granules. This allowed an image individual determination of granules independently of the slight different staining intensity of the images. By means of a median filter on the pixels obtained by this procedure, the real intensity of a pixel was substituted by the median intensity of the surrounding pixels. Based on the resulting image, the granularity was determined and the number, mean area and mean intensity of granules were calculated. Only granules of a minimum size of 5 pixels were considered in order not to include granules below the resolution limit of the instrument.

Statistical evaluations were performed with the software OriginLab's OriginPro 2022. To test the significance of the data, the two-sample *t*-test was used for normally distributed data (size of nuclei); otherwise, the Kruskal–Wallis ANOVA test was used. In addition, a distinction was made between different significance levels: in the Results Section (see Section 3), a *p*-value of $p \geq 0.05$ corresponded to "non-significant" (n.s.), $p < 0.05$ to "significant" (*), $p < 0.01$ to "highly significant" (**) and $p < 0.001$ to "very highly significant" (***). The results of the *p*-values are presented in detail in the Supplementary Materials.

## 3. Results

As described in more detail in Section 2 (Materials and Methods Section), we used a commercially available fibroblast cell line (CRL-2522) under three different cell culture conditions (normal proliferating "CRL2522run", quiescent "CRL2522dorm", proliferation reactivated "CRL2522wake"; details of the preparation of these cell forms, their functioning and their special cell culturing conditions are published elsewhere) and under irradiation treatment with doses of 2 Gy ("CRL2522_2Gy,1.5h", "CRL2522_2Gy,8h") and 4 Gy ("CRL2522_4Gy,1.5h"). In addition, the glioblastoma cell line U87 established in our laboratory and the breast cancer cell line SkBr3 (for practical reasons, we used the abbreviation SKBR3 in the text) with and without radiation treatment ("SKBR3_0Gy", "SKBR3_4Gy,1.5h") were analyzed. To study the effects of azacitidine, a mouse model cell line P19 ("P19Aza−", "P19Aza+") was used. In Figure 1, typical images representative for cells under the labeling of euchromatin (EC represented by antibodies against H3K4me3), cohesins and heterochromatin (HC represented by antibodies against H3K9me3) are shown in order to obtain a qualitative impression of the samples and effects investigated.

## Human cell lines

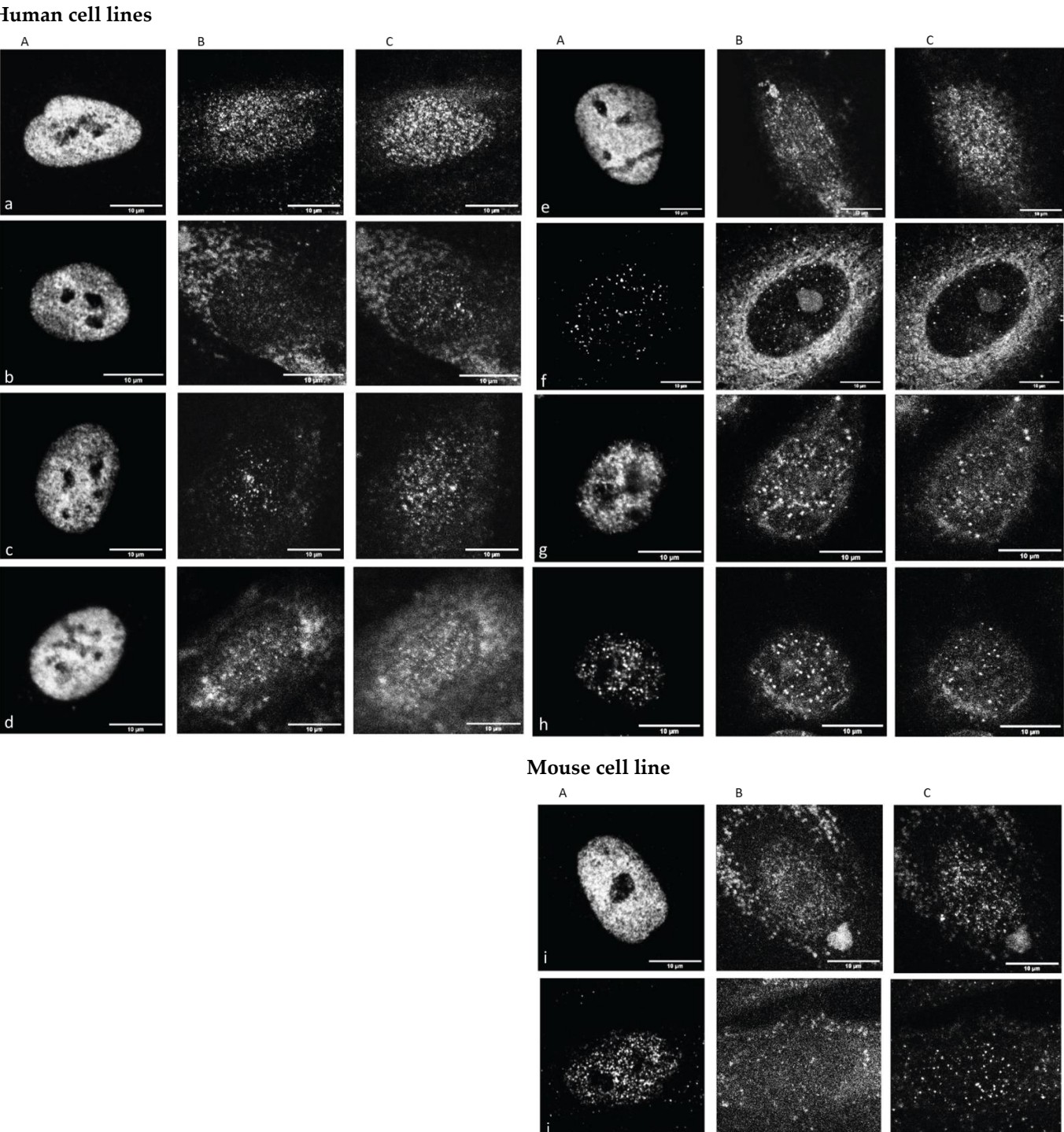

**Figure 1.** Representative confocal images of cell nuclei after H3K4me3 labeling (**A**), cohesin labeling (**B**) and H3K9me3 labeling (**C**) for fibroblasts (**a**) CRL2522run, (**b**) CRL2522dorm, (**c**) CRL2522wake, (**d**) CRL2522_2Gy,1.5h, (**e**) CRL2522_2Gy,8h, (**f**) CRL2522_4Gy,1.5h; for breast cancer cells (**g**) SKBR3_0Gy and (**h**) SKBR3_4Gy,1.5h; for mouse cells (**i**) P1 Aza- and (**j**) P19Aza+. (Note that in the cases of weak cohesin labeling (e.g., (**Bb**), (**Bc**), (**Bd**), (**Be**), (**Bf**)) or weak H3K9me3 labeling (e.g., (**Cd**), (**Ce**), (**Cg**)), the nucleus can be identified by the region of a low background supported by the DAPI mask (see Figure S1). The background signals outside the nuclei were not considered for evaluations.

The CRL2522 fibroblast nuclei showed a strong H3K4me3 signal in all untreated cases, with the exception of the quiescent one. In contrast to H3K4me3, H3K9me3 and cohesin

signaling were dispersed. For the cells subjected to 4 Gy irradiation treatment, all signal types were strongly dispersed and had to be carefully analyzed quantitatively. These highly increased H3K4me3 signals were even more intensified in U87 (data not shown in Figure 1). The irradiated SKBR3 cells and azacitidine-treated P19 cells showed a loss in H3K4me3 signaling compared to the untreated controls. These visual inspections encouraged further quantitative analyses. For those further quantitative evaluations, the cell nuclei were segmented according to the masks obtained by their individual DAPI counterstaining image (Supplementary Figure S1).

In a first approach, the sizes of the cell nuclei were quantitatively compared. It was noticeable that the tumor cells SKBR3 and U87 had, on average, significantly smaller cell nuclei than the fibroblasts CRL2522 and murine embryonic carcinoma stem cells P19 (Figure 2). The untreated CRL2522 and P19 cells both had a mean nuclear size of about 250 $\mu m^2$, whereas U87 with 136 $\mu m^2$ and SKBR3 with 157 $\mu m^2$ were significantly smaller. With regard to the CRL2522 preparations, the cells irradiated with 4 Gy showed, on the one hand, a non-significant larger mean size of about 300 $\mu m^2$ compared to the untreated cells (CRL2522run), and on the other hand, a significantly larger width of the measured values compared to the remaining CRL2522 preparations. In addition, the resting cells (CRL2522dorm) showed a strikingly high number of outliers and significantly smaller cell nuclei than the untreated cells. Both preparations irradiated with a dose of 2 Gy behaved similarly to the untreated and reactivated cells (CRL2522wake). However, CRL2522_2Gy,1.5h with a mean size of 233 $\mu m^2$ had significantly smaller nuclei than CRL2522_4Gy,1.5h (299 $\mu m^2$). In addition, irradiated SKBR3 cells had a significantly smaller mean nucleus size (about 143 $\mu m^2$) than the non-irradiated cells (about 157 $\mu m^2$). In the treatment of P19 cells with azacitidine, no significant effect on the size of the nucleus was observed. While the same irradiation therapy (4 Gy, 1.5 h repair time) for CRL2522 caused an enlargement of the cell nuclei, although not significant, it caused a significant reduction in the size of the cell nuclei for SKBR3.

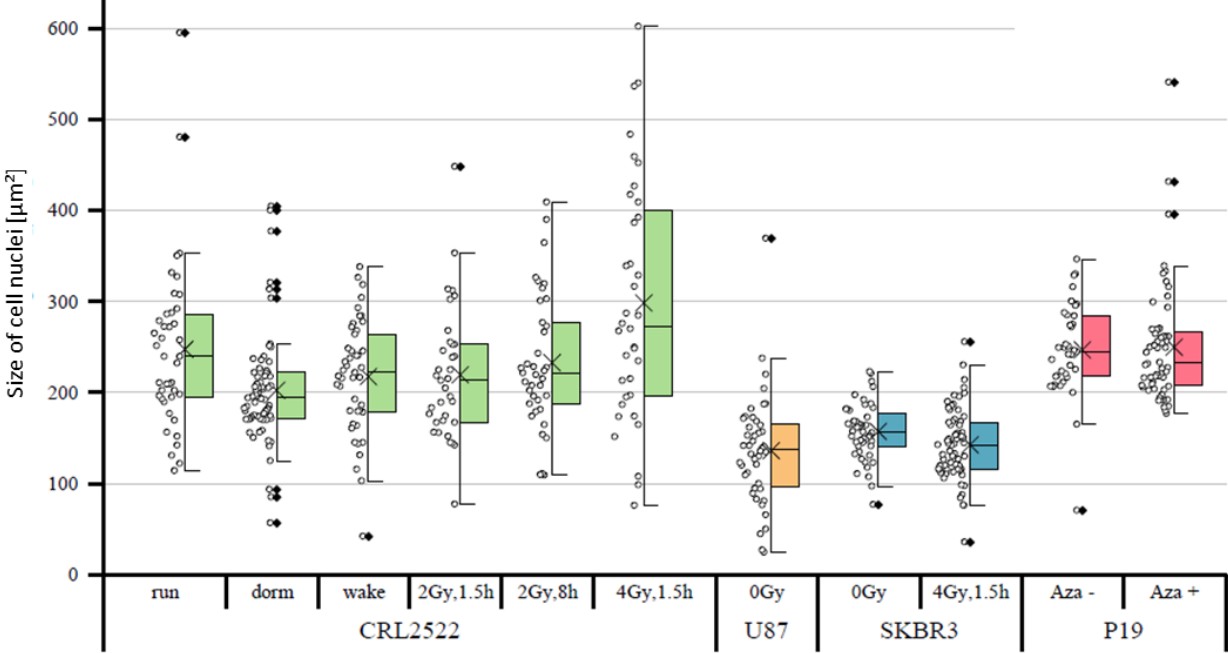

**Figure 2.** Representative datasets for the sizes of cell nuclei of the different specimens analyzed. For each dataset, the cell nuclei were selected from low-background regions of 3–5 slides after a quality check of the preparations. The boxplots show the median values (horizontal line), the lower and upper quantiles (colored boxes) and ±2 standard deviations (error bar) from the mean value (cross ×); the diamond symbols (◊) indicate the outliers. For statistical comparisons, all *p*-values can be found in the Supplementary Materials (Table S1).

In the following step, we determined the amount of H3K4me3 used as a euchromatin marker (Figure 3a), H3K9me3 used as a heterochromatin marker (Figure 3b) and cohesin (Figure 3c) by calculating the average signal intensities of the respective labels times the sizes of the cell nuclei in pixels.

If the amounts of H3K4me3 of the different untreated preparations were compared, CRL2522run and U87 had the highest amounts of H3K4me3 indicating high genetic activity. The amount of H3K4me3 in P19 preparations was significantly lower (about a quarter less), which was compatible with the different genomes of humans and mice. The SKBR3 preparations even had only about one tenth of the amount of H3K4me3 of the CRL2522run.

With regard to the CRL2522 preparations, it can be seen that normal proliferating versus quiescent and quiescent versus reactivated cells differ significantly (note the logarithmical scale in Figure 3a). There is a decrease from proliferating to quiescent and an increase from quiescent to reactivated cells, which represents the genetic activity of the different specimens; thereby, the amount of H3K4me3 in the reactivated cells nearly reaches the level of the proliferating untreated cells. With irradiation, significant changes also occurred, depending on the radiation dose. In the case of a dose of 2 Gy, there was a significant increase in H3K4me3 compared to non-irradiated normal proliferating cells. This can be assigned to the increased repair activity of these cells also at a later time point (8 h) post-irradiation.

In contrast, the amount of H3K4me3 decreased to less than one tenth 1.5 h after irradiation with a dose of 4 Gy. This may indicate that the cells irradiated with 4 Gy are so heavily damaged that they have to slow down many other genetic activities, except the necessary DNA repair activities in order to survive. This behavior was also observed for SKBR3 cells exposed to a 4 Gy radiation dose. Here, the amount of H3K4me3 was in the same range as for CRL2522 after 4 Gy radiation exposure; although, the amount of H3K4me3 in the non-irradiated SKBR3 control was less when compared to the untreated proliferating CRL2522 control.

The amount of H3K9me3 significantly differed between all the untreated preparations, except between CRL2522run and U87 (Figure 3b). The amount of H3K9me3 in these two preparations was approximately at the same level, which correlated well to the H3K4me3 data. Additionally, the SKBR3 preparations showed a reduced amount of H3K9me3 (about half, compared to CRL2522). The amount of H3K9me3 in the P19 preparation was about one third of the CRL2522run and U87 cells, which could be explained by the different species of origin of the cells. In contrast to H3K4me3, significant changes in the amount of H3K9me3 were not measured for the quiescent and reactivated CRL2522 cells. However, the statistical width of the data value distribution was less when compared to the untreated proliferating CRL2522 cells.

1.5 h after radiation exposure to 2 Gy, there was apparently more than a doubling of measured H3K9me3 found when compared to the control. This doubling was no longer present after a repair time of 8 h. This apparent increase in H3K9me3 labels shortly after irradiation with a dose of 2 Gy was well correlated to the other observations [27] where, after the induction of double-strand breaks in heterochromatin, a considerable relaxation of heterochromatin occurred and, thus, the accessibility for the labeling antibodies increased, since the methylation sites of heterochromatin labeled here were very stable, independently of the real compaction. In contrast to the irradiation with a dose of 2 Gy, a radiation exposure to 4 Gy induced more DNA double-strand breaks in the heterochromatin. This meant that there were more damage sites, which induced heterochromatin relaxation and increased the numbers of H3K9me3 labels. All these additional damage sites required additional proteins for repair so that, although the nucleus size increased, the overall average relaxation of heterochromatin (represented by H3K9me3 labeling) reduced due to space limitations for simultaneous damage repairs at the heterochromatin–euchromatin border. The relaxation of heterochromatin (represented by H3K9me3 labeling) irradiated by a dose of 4 Gy was less [28], accompanied by reduced accessibilities of the antibody constructs to the methylation sites, which could explain the observed CRL2522_4Gy,1.5h result.

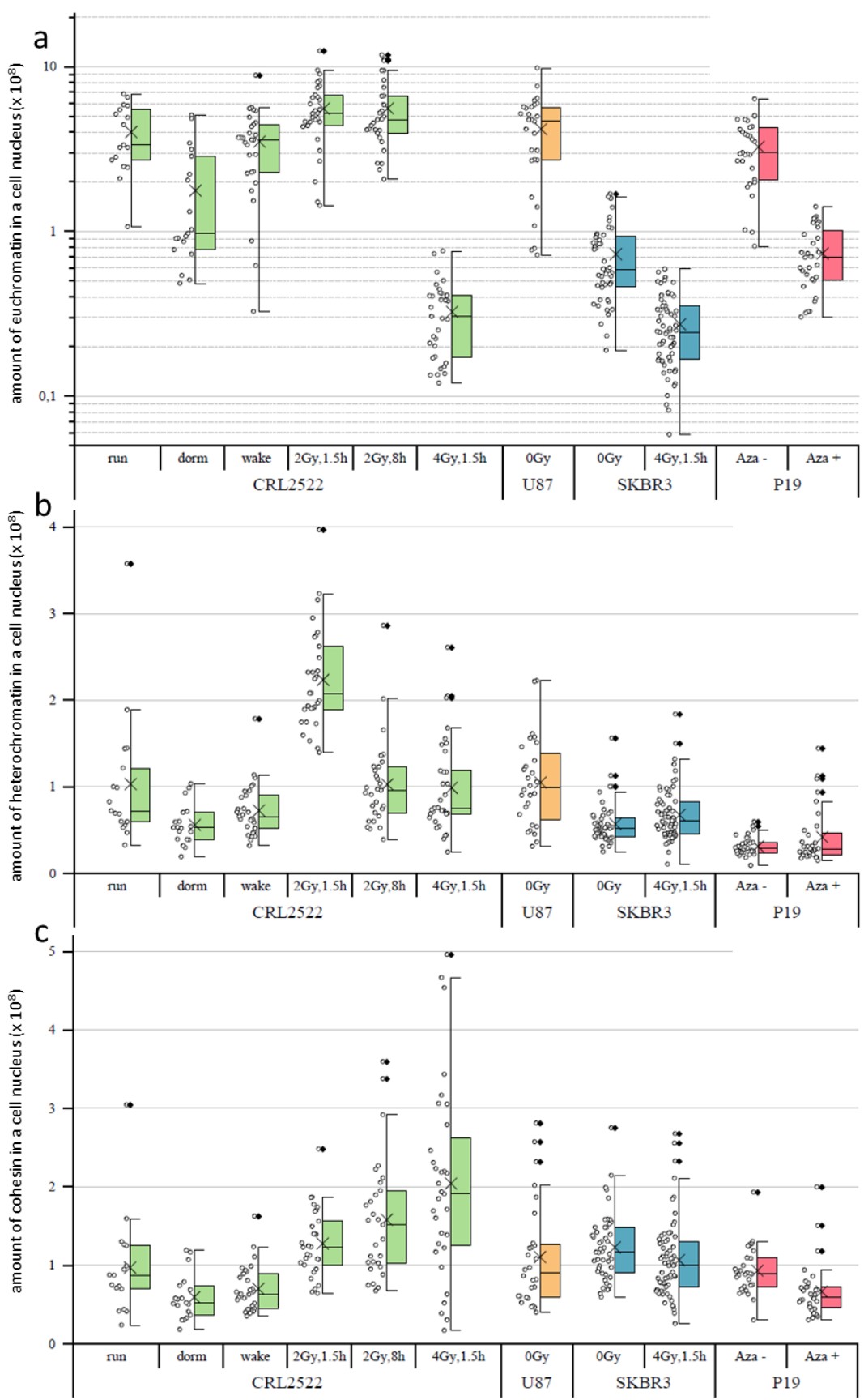

**Figure 3.** Representative datasets of the relative amounts of euchromatin (represented by H3K4me3) (**a**), heterochromatin (represented by H3K9me3) (**b**) and cohesin (**c**) of cell nuclei from the different

specimens analyzed. For each dataset, the cell nuclei were selected from low-background regions of 3–5 slides after a quality check of the preparations. The ordinate is given as the relative signal intensity of the respective label times the size of the cell nucleus in pixels. The boxplots show the median values (horizontal line), the lower and upper quantiles (colored boxes) and $\pm 2$ standard deviations (error bar) from the mean value (cross $\times$); the diamond symbols ($\lozenge$) indicate the outliers. For statistical comparisons, all *p*-values can be found in the Supplementary Materials (Tables S2–S4).

Upon the irradiation of SKBR3, a significant, but small, increase in the amount of H3K9me3 was present. The reason for that was compatible to the explanation for CRL2522_2Gy,1.5h. The treatment of P19 with azacitidine also resulted in a slight increase in the mean value of H3K9me3, which, however, was not significant.

Looking at the amount of cohesin (Figure 3c) in the untreated preparations, only the SKBR3 untreated control differed from CRL2522run and P19Aza− significantly; the latter could be explained by the different species. As with H3K4me3, there was a significant decrease in the amount of cohesin in CRL2522 from the normal proliferating to the quiescent cells, and, again, an increase from quiescent to reactivated ones. Both types of radiation exposures resulted in a significant increase in the amount of cohesin in the cells. A positive correlation between the irradiation intensity and amount of cohesin seemed to occur. CRL2522_4Gy,1.5h had a significantly higher amount of cohesin in the cell nucleus than SKBR3_4Gy,1.5h.

The azacitidine-treated P19Aza+ preparation also show a significant decrease in the amount of cohesin in the nucleus, compared to the untreated preparation.

As observed by a visual inspection, the signal distributions of H3K9me3 and cohesin always showed granularity; although, the images were not acquired under super-resolution imaging conditions [27]. This meant that the observed granules were detected in a size range larger than 250 nm (=resolution limit of diffraction limited fluorescence microscopy). Due to diffraction, the minimum sizes of the granules could not be determined. In Figure 4, the results for the granularity values of H3K9me3 (Figure 4a–c) and cohesin (Figure 4d–f) are summarized. In all cases, only granules with a size larger than five pixels were considered.

A granular distribution was observed for heterochromatin represented by H3K9me3 (Figure 4a–c). As already expected from the images (see Figure 1), on the basis of the qualitative evaluation, in the preparation of CRL2522_2Gy,1.5h, there was hardly any granulation of H3K9me3 observed. After a quantitative evaluation, we excluded these specimens from further comparisons. In the case of P19, some of the recorded cell nuclei were rejected, as they either had a signal that was too weak or no granulation.

Comparing the untreated cells with each other, it was noticeable that U87, with a mean of about 32 and a median of 22 granules per nucleus, had a significantly higher number of H3K9me3 granules than the other preparations (Figure 4a). In addition, the data points were significantly more widely scattered. The other three untreated preparations did not differ significantly from each other and all had a comparably low median value between 6 and 9. Taking into account that CRL2522_2Gy,1.5h did not show sufficient granulation, it could be observed that, after an 8 h repair time, granulation occurred again. This was compatible with heterochromatin relaxation and recovery after irradiation with 2 Gy. It was noticeable that the data of CRL2522_2Gy,8h had a wider scatter than the rest, indicating that granulation was just in progress and the cells were in different stages.

No significant change occurred with SKBR3 either, as with P19. Here, a non-significant increase was observed for the mean number of H3K9me3 granules from about 9 for the untreated to 12 for the cells treated with azacitidine.

We also determined the size of the granules. With an average granule size of about 20 pixels, SKBR3_0Gy had larger granules than all the other untreated preparations. With about 16 pixels as the mean value, U87 lies between the other preparations. Comparing the CRL2522 preparations with each other, no significant change was observed between CRL2522run, CRL2522dorm and CRL2522wake. The irradiation of the fibroblasts with 4 Gy led to a significant enlargement (about a factor of 2) of the H3K9me3 granules. It was

observed that, at 8 h after irradiation with 2 Gy, the granule size was again approximately at the level of the untreated cells.

For the cells of SKBR3 irradiated with 4 Gy, there was also an enlargement of the granules observed; however, this was not significant. A significant enlargement of the granules by a factor of approximately two occurred when P19 was treated with azacitidine. Additionally, SKBR3 cells irradiated with 4 Gy significantly differed from CRL2522 cells irradiated with 4 Gy. The mean value of CRL2522 was higher than that of SKBR3.

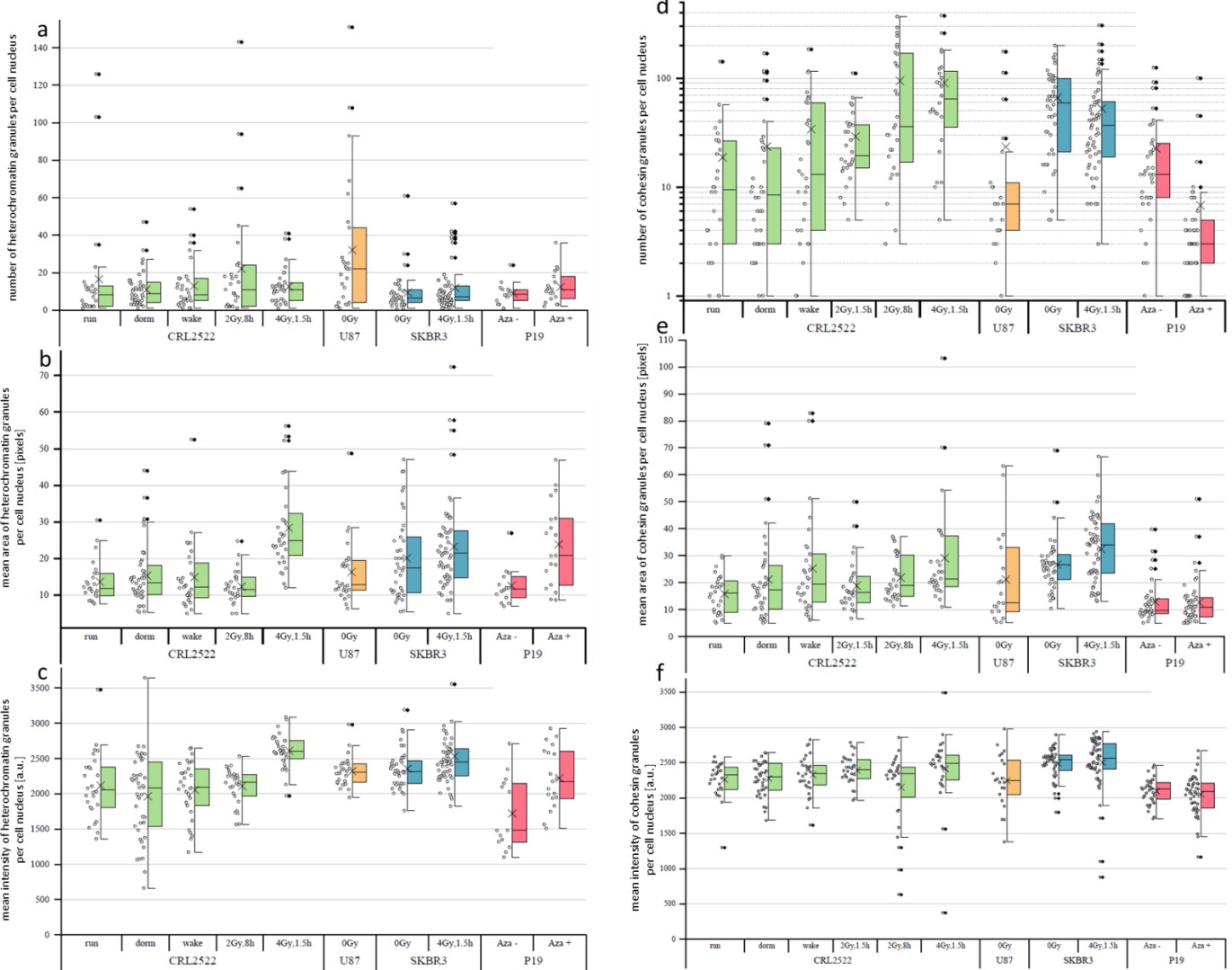

**Figure 4.** Representative datasets of the granularities of (**a**–**c**) heterochromatin (represented by H3K9me3) and (**d**–**f**) cohesin of cell nuclei of the different specimens analyzed. For each dataset, the cell nuclei were selected from low-background regions of 3–5 slides after a quality check of the preparations. The ordinate is given as the number (**a**,**d**), mean area (**b**,**e**) and mean intensity (**c**,**f**) of the respective granules of a cell nucleus. The boxplots show the median values (horizontal line), the lower and upper quantiles (colored boxes) and ±2 standard deviations (error bar) from the mean value (cross ×); the diamond symbols (◊) indicate the outliers. Note the logarithmic scaling of the ordinate in (**d**). For statistical comparisons, all *p*-values can be found in the Supplementary Materials (Tables S5–S10).

For the intensities, U87 and the preparations of CRL2522 and SKBR3 showed approximately the same values, except for the preparations irradiated with 4 Gy. The latter exhibited a significant increase in intensity and had the highest granule intensity of all the preparations, with a mean value of about 2600. The azacitidine-treated P19 cells also showed a significant increase in the intensity levels of the H3K9me3 granules.

We evaluated the granulation of cohesin (Figure 4d–f). In the untreated preparations, SKBR3, with 66 granules per nucleus, stood out significantly from the other preparations by a factor of three. The median of 59.5 was at least six-times higher (note that Figure 4d presents logarithmic scaling). Analyzing the CRL2522 preparations, CRL2522dorm and CRL2522wake did not show any significant deviations from CRL2522run. The irradiated preparations CRL2522_2Gy,1.5h and CRL2522_4Gy,1.5h, however, deviated significantly from each other and from CRL2522run. CRL2522_4Gy,1.5h had the largest mean value of 64 granules per nucleus, the largest mean value of the three preparations, and a median value of about 90 granules per nucleus. CRL2522_2Gy,1.5h was between CRL2522_4Gy,1.5h and CRL2522run.

In the breast cancer cell line SKBR3, there was no correlation between irradiation and an increase in the number of granules. Here, no significant deviation was observed between the untreated and irradiated cells. Both the median and mean values for the irradiated cells differed by about 22 (median)- and 13 (mean).

As already observed, the number of cohesin granules decreased significantly from untreated P19Aza− to azacitidine-treated P19Aza+ cells. While the median values for the untreated cells was 13 and the mean value was about 23 granules per nucleus, there were hardly any granules left in P19Aza+. This could be seen especially for the median with a value of 3 granules per nucleus. When looking at CRL2522_4Gy,1.5h and SKBR3_4Gy,1.5h, it was noticeable that the former had a significantly higher number of granules than the latter; although, the opposite was true in the case of the untreated preparations of the cells.

In addition to the number of cohesin granules, the size of the granules could also be important. One data point in Figure 4e corresponds to the mean size of the cohesin granules of a nucleus in the pixels. For the size of the cohesin granules in the nuclei of the untreated cells, only SKBR3_0Gy had a significantly higher value, with a mean value of about 27 pixels. P19Aza− had the lowest mean value of about 13 pixels. Furthermore, U87 showed a very broad and asymmetrical scattering of values.

For CRL2522, a slight increase of about 2 pixels in the median was observed from CRL2522run to CRL2522dorm and from CRL2522dorm to CRL2522wake; however, this was not significant. Comparing CRL2522run with CRL2522_4Gy,1.5h as well as CRL2522_2Gy,1.5h with CRL2522_4Gy,1.5h, a significant enlargement of the granules was observed. Although there was a tendency to decrease the number of cohesin granules from SKBR3_0Gy to SKBR3_4Gy,1.5h, there was a significant increase in the mean size of these granules from about 27 to 32 pixels.

Between the two preparations of P19, no significant differences were observed in size. Both had a mean value of about 13 pixels.

In Figure 4f, one data point corresponds to the mean intensity of all pixels of a nucleus (each pixel has an intensity value between 0 and 4095), which are assigned to cohesin granules and not the mean intensity of the granules. Each pixel is thus equally weighted, regardless of whether it is in large or small granules. In the untreated cells, no significant difference was observed, except between U87 and CRL2522. All other combinations differed significantly from each other.

P19, with a mean value of about 2100, had the lowest mean value and SKBR3, with a mean value of about 2490, had the highest intensity. CRL2522 and U87 were similar to each other with mean values of about 2260 and 2250, respectively.

There were no relevant differences between CRL2522run, CRL2522dorm and CRL2522wake. From CRL2522run to CRL2522_4Gy,1.5h, there were significant changes. The intensity increased for the irradiated preparations.

No significant differences were found for P19 or SKBR3. Additionally, between the two preparations of SKBR3 and CRL2522 irradiated with 4 Gy, no significant difference was observed.

## 4. Discussion

In this article, the results of single cell experiments considering the microscopic outcomes of the specific labeling of H3K9me3 (representing heterochromatin), H3K4me3 (representing euchromatin) and cohesin were summarized and given a statistical description. A representative selection of the cell lines was analyzed. Some of these cells were subjected to typical cancer treatments, i.e., radiation or azacitidine. Although only well-established, standardized tools of a computer image analysis program were used, which ensured the high reproducibility of the evaluation process, the cells in most cases of the cell lines analyzed showed a broad variability in the quantitative values. Using standard box plot graphics allowed for the description of the major differences between the cell types and treatments. The calculation of the *p*-values (see Supplementary Materials) provided hints about the quality of significance of these differences. Nevertheless, it might be apposite to ask for reasons for this variability. Initially, one might think about the preparation effects that, on one hand, can never be excluded completely. However, the specimens were prepared by a well-skilled person in several replicates and the cells were selected without clearly visible differences in the quality. Therefore, on the other hand, real biological reasons had to be included in the consideration. Recently, Alekseenko et al. [29] pointed out the reasons for the fundamentally low reproducibility of the quantitative results between individual cancer cells. For each cell that can be subjected to single cell experiments, slightly different conditions in the microenvironment might lead to small functional differences in the individual cells; although, they all produce the same tumor or cell type. This might be negligible in bulk experiments; however, in single cell measurements, it can become more prominent. In other words, the more precisely single cell measurements are performed, the more individual variations may become obvious.

Another more general aspect that may be considered is the specificity of the antibodies used to label euchromatin (H3K4me3) and heterochromatin (H3K9me3). Both antibodies are known to show some cross-labeling of non-euchromatic and non-heterochromatic regions, respectively. But according to our experience this seems to be negligible in experiments as being presented here. There are other antibodies against euchromatin and heterochromatin available; however, none of them are free from such cross-labeling. Detailed studies of established databases induced the decision to use these antibodies for many of our investigations (see [9,11,16] and citations therein), since it seemed that these antibodies were the best compromise between labeling specificity and labeling efficiency. In order to keep the presented results compatible and comparable to many other results we published in the last decades, have decided to continue to use these antibodies.

In the following section, the discussion is separated into sub-chapters for a better overview. As mentioned in the Materials and Methods Section (see Section 2.2), it was our intention to consider the whole chromatin response under different aspects and to compare the appropriate sub-groups of our datasets accordingly. The results in general should be seen within the frame of other investigations conducted using super-resolution microscopy, e.g., single-molecule localization microscopy. These investigations were focused on repair foci analyses (γH2AX, 53BP1, Mre11, Rad51, etc.), which were not considered further here (see [9,30,31] and citations therein).

### 4.1. Comparison of CRL2522run, U87, SKBR3_0Gy and P19Aza−

When looking at the untreated preparations of the different cell lines, it was noticeable with regard to the nucleus sizes that CRL2522run and P19Aza− were about the same size; although, the genome sizes in the base pairs (bps) were quite different (2600 Mbp mouse vs. 3200 Mbp human) [32]. Nuclei of SKBR3_0Gy and U87, however, were significantly smaller; thereby it was observed that fibroblasts could be up to 50 µm in size [33] and were

about twice as large as the average human cell [34]. The difference in the mean sizes of the untreated preparations were not unusual since there were generally more significant differences between the cell sizes of different cell types within an organism than between different mammalian species generally [35].

The amounts of H3K4me3 (mostly representing euchromatin) and H3K9me3 (mostly representing heterochromatin) were significantly different between most of these four preparations. For the different species, human and mouse, this can be explained by the species-specific activities of the genome and genome size. In humans, only between CRL2522run and U87, no significant difference was observed. SKBR3_0Gy had by far the lowest amount of H3K4me3. This may be mainly due to the fact that it is distributed over a smaller area in general than in the other preparations. On the other hand, in breast cancer cells, there is often an up-regulation of the protein PRC2, which is responsible for the modification H3K27me3, which marks the facultative heterochromatin [36,37]. This leads to an increase in the amount of facultative heterochromatin and, consequently, a decrease in the amount of H3K4me3-positive euchromatin.

The amount of cohesin showed a significantly smaller discrepancy between the four preparations. Only SKBR3_0Gy differed from CRL2522run and P19Aza−. The increase in cohesin in SKBR3_0Gy may be correlated to the increase in facultative heterochromatin, as mentioned above.

In general, different cells sometimes greatly differ in their gene regulation. Since all stained structures were part of gene regulation, the differences observed were probably due to this effect.

The amounts of cohesin and H3K4me3 or H3K9me3 did not correlate in principle. SKBR3_0Gy, for example, had significantly less H3K4me3-positive euchromatin and H3K9me3-positive heterochromatin than U87; however, the amount of cohesin was not significantly less; it even showed a tendency towards a higher amount. Since H3K9me3 marks the constitutive heterochromatin and this differs slightly from cell type to cell type within a species, the significant differences in the amount of H3K9me3 between SKBR3 and the other two human cell lines are not self-explanatory. However, cancer cells behave differently in regulatory processes than healthy cells; it might be assumed that the constitutive heterochromatin is differently organized [11] and thus has different accessibility for the labeling antibodies, which can ultimately explain these differences.

There were no significant differences between U87 and CRL2522run in the number, size and intensity of cohesin granulation, while SKBR3 _0Gy was found to be higher than the values of these two cell lines. The number of H3K9me3 granules in U87 was significantly higher than in the other three preparations, which did not significantly differ from each other. In addition, the scatter within the U87 values was also significantly greater, which may have been due to irregularities in either the staining of this preparation or in the H3K9me3-positive heterochromatin configuration of this cell line.

In terms of the granule size, on the other hand, it was the SKBR3_0Gy preparation that stood out significantly from CRL2522run. It seems that CRL2522run cells have a constant H3K9me-positive heterochromatin configuration from which the cancer cells differ in individual points. All differences observed in the cancer cells were probably due to the different regulatory processes and hypo-methylation described in the Section 1. This may support the argument that regulatory processes in the genome and chromatin organization closely interact.

*4.2. Comparison of CRL2522run, CRL252dorm and CRL2522wake*

A significant reduction in the cell nucleus size was observed from CRL2522run to CRL2522dorm, which was reconstituted in CRL2522wake. This reduction in the nucleus size was accompanied by a considerable reduction in H3K4me3 or a strong hypo-methylation of H3K4me3, respectively, which was reported for quiescent cells [38]. In addition, quiescent cells also show a hypo-methylation of H3K9me3 (our heterochromatin marker) [38], which may explain our results for CRL2522dorm.

Euchromatin is a loosely packed chromatin form and usually occupies a larger space then the same amount of heterochromatin. In resting cells with these hypo-methylations, especially hypo-trimethylations, both chromatin forms seem to be in some intermediate form of condensation, which results in a generally relaxed chromatin organization that might also offer many opportunities for re-programmability [38]. In the shift from CRL2522dorm to CRL2522wake, quiescence should be fully reversible if the original culturing conditions are refreshed. In the case of H3K4me3 and H3K9me signals, significant increases were observed, which meant that the "wake state" did not differ from the run state. Additionally, the size of the cell nuclei increased to the level of the CRL2522run cells again. Additionally, the amount of cohesin present in the nucleus seemed to follow this recovery effect.

When considering cohesin granulation, the observations of restoration did not apply. Here, there were hardly any highly significant differences between these three preparations with regard to the number, size and intensity of the granules. Only CRL2522wake had significantly more granules than CRL2522dorm indicating that, with the recovery of the genome function, a recovery of chromatin organization occurred. In terms of the granule size, there was a tendency towards an increase from CRL2522run to CRL2522dorm to CRL2522wake; however, this was not significant. The increase in the amount of cohesin from CRL2522dorm to CRL2522wake was at least partly caused by the increase in cohesin in the granules. The decrease in the amount of cohesin from CRL2522run to CRL2522dorm could not be associated with a decrease or reduction in the number of cohesin granules. This may only be explained by suggesting that cohesin was not present in the granules.

Considering that CRL2522dorm cells are resting cells and thus presumably show less genome activity, it is also conceivable that less cohesin is synthesized for TAD formation. Looking at cohesin granularity, it also seems that CRL2522wake still possesses some configuration of CRL2522dorm. This means that the total amount of cohesin in the nucleus may be restored, but not its distribution. The reasons for this might be a type of "cell memory" or that the restoration of the granulation was a slow biological process that was incomplete when the cells were subjected to microscopy.

The granulation of H3K9me3 did not differ significantly in any aspect in these three preparations. This may be justified by the fact that the constitutive heterochromatin in each cell is composed of approximately the same genome segments. The contrast to the variation in the total amount of H3K9me3 signals (due to hypo-methylation of heterochromatin) suggests that this variation in methylation may occur in the heterochromatin, which is not organized into granules.

### 4.3. Comparison of CRL2522run, CRL252_2Gy,1.5h and CRL2522_4Gy,1.5h

The nuclear sizes of the two irradiated preparations did not differ significantly from run. However, the mean value of the cells irradiated with 4 Gy was significantly higher than that of the cells irradiated with 2 Gy. This could have been due to the increased DNA repair processes that occurred in CRL2522_4Gy,1.5h cells compared to CRL2522_2Gy,1.5h, since the compact structures in heterochromatin regions must relax before the double-strand break can be repaired [39]. As a consequence, heterochromatin together with protein machinery for repair takes up more space.

The amount of H3K4me3 increases considerably from CRL2522run to CRL2522_2Gy,1.5h, which may be correlated to the increased repair activity. However, the H3K4me3 amount decreased considerably for CRL2522_4Gy,1.5h. This was in agreement with Seiler et al. who observed an active removal of the histone modification H3K4me3 after the irradiation of different cell lines (HeLa, U2OS and hTERT-immortalized BJ1) [40]. Accordingly, this should also be the case at the lower radiation dose as Seiler et al. also analyzed cells treated with a radiation dose of 2 Gy.

This was not observed in our work. Additionally, the amount of H3K9me3 increased significantly from CRL2522run to CRL2522_2Gy,1.5h, whereas for CRL2522_ 4Gy,1.5h, it remained about the same. Seiler et al. [40] also found such an increase in heterochromatin in irradiated cells; however, this was facultative heterochromatin, and therefore the histone modification H3K27me3 increased and not H3K9me3 as in the present study. The amount of H3K9me3 should not undergo any changes [40,41], which is consistent with the data from the fibroblasts irradiated with 4 Gy (CRL2522_4Gy,1.5h), but not with those fibroblasts irradiated with 2 Gy (CRL2522_2Gy,1.5h). However, it was shown that heterochromatin was more relaxed after a DSB induction at lower doses than at 4 Gy [28]. This greater relaxation may increase the accessibility of the H3K9me3 methylation sites for the labeling antibodies, which can explain the detected differences between CRL2522_2Gy,1.5h and CRL2522_4Gy,1.5h.

Furthermore, it was noticeable that the amount of cohesin increased with the irradiation of the cell compared to CRL2522run. A reason for this could be the increased demand for cohesin during repair after radiation treatment. This is presumably due to the fact that, in the DNA repair mechanism of homologous recombination, cohesin is required to bring the two sister chromatids into close spatial proximity to each other so that the DSBs created by the irradiation can be repaired [42,43]. Since the cells used were not synchronized, it can be assumed that some of the cells were in the S or G2 phases during irradiation so that repair by homologous recombination was possible. Since this explanation may be reasonable for CRL2522_2Gy,1.5h, it is not initially true for CRL2522_2Gy,8h. At this time point, the repair activity should be reduced or even completed, a contradiction to a further increase in cohesin. However, the decrease in cohesin again needs protein machinery that takes some time to be available, which might explain the delayed existence of cohesin. Moreover, if many of the cohesin molecules are assumed to be free after 8 h and therefore better accessible for the antibodies, the additional increase in cohesin compared to CRL2522_2Gy,1.5h may become reasonable. However, this can be the subject of further investigations.

The significant increases in the number of cohesin granules with an increasing irradiation dose might be related to the number of DSBs induced and in need of repair. Furthermore, there is a significant increase in the granule size and intensity compared to CRL2522run. This might be due to the fact that cohesin recruited to several DSBs in close proximity to each other forms apparently enlarged granules instead of several distinct granules because of the point spreading due to the diffraction in the microscope used.

### 4.4. Comparison of CRL2522run, CRL252_2Gy,1.5h and CRL2522_2Gy,8h

The nucleus sizes of these preparations do not differ significantly. The amounts of H3K4me3 and H3K9me3 of CRL2522_2Gy,8h do not differ significantly from the respective amounts of CRL2522run, which is in accordance with the literature in the case of H3K9me3 in contrast to 2Gy,1.5h [40]. However, the amount of euchromatin should undergo a reduction, as an active removal of the histone modification H3K4me3 is observed after irradiation [40]. For the amount of cohesin, a non-significant tendency to increase at a repair time of 8 h compared to 1.5 h was observed. With regard to cohesin granulation, a further significant increase in the number of granules and a non-significant increase in the granule size were observed after 8 h. A possible reason for this may be that the cohesin recruited for repair has not yet been degraded.

### 4.5. Comparison of SKBR3_0Gy and SKBR3_4Gy,1.5h

These two preparations differed in the nucleus size, which strongly correlated to the amount of H3K4me3. There was also an increase in the amount of H3K9me3. The amount of cohesin showed a significant decrease. The reduction in the amount of cohesin was also accompanied by a significant reduction in the number of cohesin granules and a significant increase in size. This means that there may also be a partial rearrangement of cohesin in the cell nucleus. This may be related to the role of cohesin in the repair of DSBs described above. With regard to the granulation of H3K9me3, no significant differences were shown, either

in the number and size of the granules or in their intensity. This was the case, even though there was an increase in H3K9me3 after irradiation, which contradicted the literature but could also explained by the improved accessibility of the labeling sites.

*4.6. Comparison of SKBR3_4Gy,1.5h and CRL2522_4Gy,1.5h*

The two similarly treated preparations of SKBR3 and CRL2522 cells differed significantly in the sizes of their nuclei. SKBR3 had smaller cell nuclei than CRL2522. This difference was mainly due to the different cell types. Both preparations showed a decrease in the amount of H3K4me3 after irradiation. However, CRL2522 4Gy,1.5h showed a greater decrease compared to CRL2522run as SKBR_4Gy,1.5h compared to SKBR3_0Gy. In terms of the amount of cohesin, there was a significant increase for CRL2522_4Gy,1.5h and a significant decrease for SKBR3_4Gy,1.5h compared to the untreated cells. This may indicate homologous recombination repair processes in CRL2522, but not in SKBR3. The different behavior of the cells with regard to the amount of cohesin was also reflected in the cohesin granulation.

While CRL2522_4Gy,1.5h showed an increase in the number of granules, SKBR3_4Gy,1.5h showed a decrease. The size, on the other hand, showed a similar increase in both preparations after irradiation; thereby SKBR3 had significantly larger granules already before irradiation. Thus, after irradiation, different repair reactions may be observed in the two cell types with regard to cohesin granulation.

There was no significant difference in the number of H3K9me3 granules. However, there was a greater increase in granule size and intensity of H3K9me3 in CRL2522_4Gy,1.5h compared to the untreated cells than in SKBR3_4Gy,1.5h. Again, this may indicate the different reactions during repair. This might be due to radio-resistance of the cell types. CRL2522 is less radio-sensitive than SKBR3.

It should also be mentioned that by considering the results for H3K4me3 and H3K9me3 in combination, apoptotic effects, especially early apoptosis, cannot be completely excluded; although, previous experiments using these cell types indicate a negligible apoptosis rate after radiation treatment.

In general, it can be stated that the two cell types react differently to irradiation. CRL2522 seems to be more greatly damaged by radiation (larger radio-sensitivity known for fibroblasts [44]) than SKBR3. A possible explanation for this might be that the SKBR3 cell line contains a dysfunctional tumor suppressor p53 with inactivating point mutations in the gene [45]. If this transcription factor is defective or almost completely absent, it can lead to reduced DNA repair activity.

*4.7. Comparison of P19Aza− and P19Aza+*

In order to show that whole chromatin reactions are general effects of the adaptive self-organization of chromatin, we included the data of a mouse cell line and exposed them to a chemical treatment well established in human cancer therapy.

The two preparations of the murine embryonal terato-carcinoma cell line did not differ in nucleus size, but in nucleus morphology. The nuclei of the cells treated with azacitidine showed more bulges and irregularities than those of the untreated cells. The reason for this could be the disturbed gene regulation caused by the demethylation of DNA, which could also cause morphological changes [46]. The decrease in the amount of cohesin was mainly reflected in the number of cohesin granules. This was significantly lower for P19Aza+ than for P19Aza−, whereas the intensity and size of the granules did not differ significantly.

In contrast to the total amount of H3K9me3, which seemed to remain constant in our experiments, the H3K9me3 granulation showed an increase in size, intensity and number; although, the latter was not significant. This observation of the total amount of H3K9me3-represented chromatin might contradict the results of Komashko and Farnham [7], who, for the human cell lines HEK-293 (embryonic kidney cells), MCF7 (breast cancer cells), HepG2 (liver cancer cells) and HCT116 (colon cancer cells), found a general increase in the amount of heterochromatin in cells treated with azacitidine. One explanation might be the different

species of human and mouse used in the experiments. On the other hand, our results for granule formation indicate that the distribution of H3K9me3 in a cell nucleus might also be relevant after azaciditine treatment, which might have, in general, an influence on the detailed data evaluation. The analysis of different human cell lines and different chemo-therapeutics will be the task of future investigations.

## 5. Conclusions

In this article, we demonstrated that the quantification of fluorescence microscopy images of H3K9me3 (mostly representing heterochromatin), H3K4me3 (mostly representing euchromatin) and cohesin in different types of cell nuclei and under different types of anti-cancer treatment reflected the functional cell response. The results support the general hypothesis that chromatin is an adaptive self-organizing system and thus changes in the molecular compounds induce the re-organization of chromatin as a whole and change the accessibility of chromatin regions and/or the spatial neighborhood of given chromatin domains. Here, we focused on the changes in the microscale, i.e., the obtained information was the prominent changes in chromatin organization neglecting biological fluctuations that usually overlayed whole organization and re-organization processes. The following main features were detected:

(1)	The size of the nuclei strongly depended on the cell type and correlated with the amount of H3K4me3 (mostly representing euchromatin). Additionally, the number of cohesins depended on the cell type.
(2)	Quietness of gene activity was associated with size reduction and reductions in H3K4me3, H3K9me3 and cohesin. This was reversible by re-activation.
(3)	Radiation treatment increased the size variations of the nuclei and variations of cohesins.
(4)	4 Gy radiation treatment reduced the amount of H3K9me3 labeling and increased its granularity.
(5)	Except for granularity, the effects of azacitidine treatment were compatible with the effects of irradiation.
(6)	Cohesin granularity was associated with the induction of DNA damage and thus with dose.

Nevertheless, our approach still presented some shortcomings that could be overcome by improving the microscopic resolution with systems, such as STED-microscopy (Stimulated Emission Depletion) [30] or SMLM (Single Molecule Localization Microscopy) [31]. In both cases, the experimental and evaluation efforts improved and produced very detailed information about chromatin organization and its network, such as topology, from which the major function impacting re-organization behavior must be solved. Novel mathematic approaches applied on pointillist SMLM data have therefore been developed recently [47], which might provide new insights into chromatin organization and the roles of genome activities and functioning [16].

**Supplementary Materials:** The following supporting information can be downloaded at: https://www.mdpi.com/article/10.3390/cimb45100515/s1, Figure S1: representative confocal images of cell nuclei showing the application of the DAPI mask; Table S1: *p*-values (*t*-test) for comparison of the size of the nuclei; Table S2: *p*-values (ANOVA test) for comparison of the amount of euchromatin in the nuclei; Table S3: *p*-values (ANOVA test) for comparison of the amount of heterochromatin in the nuclei; Table S4: *p*-values (ANOVA test) for comparison of the amount of cohesin in the nuclei; Table S5: *p*-values (ANOVA test) for comparison of the amount of heterochromatin granula in the nuclei; Table S6: *p*-values (ANOVA test) for comparison of the mean area of heterochromatin granula in the nuclei; Table S7: *p*-values (ANOVA test) for comparison of the mean intensity of heterochromatin granula in the nuclei; Table S8: *p*-values (ANOVA test) for comparison of the amount of cohesin granula in the nuclei; Table S9: *p*-values (ANOVA test) for comparison of the mean area of cohesin granula in the nuclei; Table S10: *p*-values (ANOVA test) for comparison of the mean intensity of cohestin granula in the nuclei.

**Author Contributions:** Conceptualization, G.P. and E.F.F.; methodology, G.P. and M.H.; software, E.F.F.; validation, E.F.F., G.P. and M.H.; formal analysis, E.F.F.; investigation, E.F.F.; resources, M.H.; data curation, E.F.F.; writing—original draft preparation, E.F.F. and M.H.; writing—review and editing, E.F.F., G.P. and M.H.; visualization, E.F.F.; supervision, M.H.; project administration, M.H.; funding acquisition, M.H. All authors have read and agreed to the published version of the manuscript.

**Funding:** This research was funded by the German Federal Ministry of Education and Research (BMBF; grant FKZ: 02 NUK 058A) to Michael Hausmann.

**Institutional Review Board Statement:** Not applicable.

**Informed Consent Statement:** Not applicable.

**Data Availability Statement:** Data are available on request to the authors.

**Conflicts of Interest:** The authors declare no conflict of interest.

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
