# Peer review of "Microscopic Analysis of Heterochromatin, Euchromatin and Cohesin in Cancer Cell Models and under Anti-Cancer Treatment"

_cimb, doi:10.3390/cimb45100515_

Round 1
Reviewer 1 Report
The manuscript by Fischer et al. reports results of microscopic analysis of euchromatin, heterochromatin and cohesin expression in human fibroblasts and various cancer cells grown in different conditions or treated with X-rays or azacytidine. Microscopic analyses of euchromatin and heterochromatin were carried out with antibodies against specific histone H3 modifications, i.e. H3K4me3 and H3K9me3, respectively. In addition to the intensity of individual antibody signals, the authors also analyzed number, size and intensity of heterochromatin and cohesin granules. Reports like this could be valuable for providing primary information on dynamic changes in gross chromatin structure depending on culture conditions and treatment. Here are a few comments that should be addressed by the authors properly.
1. A rebuttal on the matter of repetition of the experiments was found in the ‘Non-published Material’. I understand the reviewer’s concern on the matter. In order to avoid the confusion, it should be clearly stated in the figure legends that the data shown is ‘a representative’ result of certain number of repetition.
2. In addition, modifications of histone H3 targeted here are not exclusively associated with euchromatin and heterochromatin. So, the authors should take caution in using the terms interchangeable with H3K4me3- or H3K9me3-positivities.
3. Punctate signals outside the nucleus in Figure 1Bb and 1Bc and unclear nuclear boundaries in Figure 1Bd, 1Cd, 1Bf and 1Cf were shown, but not explained clearly. Figures showing antibody-specific immunofluorescent signals overlapped with DAPI signals should be provided either in the main body or in the supplementary materials.
4. Is there any way to normalize the signal intensity determined in Figure 3?
5. Regarding to the notion in lines 241-247, is there any chance that the changes shown in 4 Gy- and azacytidine-treated groups (ex, decreased euchromatin and increased mean intensity of heterochromatin granules) could result from apoptotic cell death?
6 ‘cohesion’ should be changed into ‘cohesin’ in many places.
Author Response
ad Reviewer 1
The manuscript by Fischer et al. reports results of microscopic analysis of euchromatin, heterochromatin and cohesin expression in human fibroblasts and various cancer cells grown in different conditions or treated with X-rays or azacytidine. Microscopic analyses of euchromatin and heterochromatin were carried out with antibodies against specific histone H3 modifications, i.e. H3K4me3 and H3K9me3, respectively. In addition to the intensity of individual antibody signals, the authors also analyzed number, size and intensity of heterochromatin and cohesin granules. Reports like this could be valuable for providing primary information on dynamic changes in gross chromatin structure depending on culture conditions and treatment.
The authors thank the reviewer for his work and his explanation about the usefulness in research.
Here are a few comments that should be addressed by the authors properly.
- A rebuttal on the matter of repetition of the experiments was found in the ‘Non-published Material’. I understand the reviewer’s concern on the matter. In order to avoid the confusion, it should be clearly stated in the figure legends that the data shown is ‘a representative’ result of certain number of repetition.
Thank you for your understanding and the suggestion how to explain the selection. We have added appropriate comments to the figure legends and to chapter 4.2 in Materials and Methods.
- In addition, modifications of histone H3 targeted here are not exclusively associated with euchromatin and heterochromatin. So, the authors should take caution in using the terms interchangeable with H3K4me3- or H3K9me3-positivities.
Throughout the whole text euchromatin was either exchanged by H3K4me3 positive regions or it was explained that we define euchromatin by H3K4me3. The same was done for heterochromatin and H3K9me3, respectively.
- Punctate signals outside the nucleus in Figure 1Bb and 1Bc and unclear nuclear boundaries in Figure 1Bd, 1Cd, 1Bf and 1Cf were shown, but not explained clearly. Figures showing antibody-specific immunofluorescent signals overlapped with DAPI signals should be provided either in the main body or in the supplementary materials.
An explanation is given in the figure legend and examples for application of the DAPI mask are shown in the supplementary Figure S1.
- Is there any way to normalize the signal intensity determined in Figure 3?
This is an interesting question we also discussed in our group. We did not really find a solution that would give us more or even better information. If we would normalize each figure independently from each other the problem that the H3K4me3 signal covers several orders of magnitude would not change. If we would normalize all figures equally the clearness of the distributions was reduced. This finally reasoned that we were choosing the same ordinate (108) for all distributions.
- Regarding to the notion in lines 241-247, is there any chance that the changes shown in 4 Gy- and azacytidine-treated groups (ex, decreased euchromatin and increased mean intensity of heterochromatin granules) could result from apoptotic cell death?
We did not use an apoptosis marker in these experiments.
But several points could be mentioned that may indicate that after irradiation the effects of apoptosis should not dominate in euchromatin: a) from other experiments with SkBr3 we know that they very well tolerate 4 Gy with very low (negligible) apoptosis rate. b) SkBr3 is more radio-resistant than fibroblasts. If in the case of fibroblasts significant apoptosis would occur, we would not expect the same behaviour as for SkBr3.
On the other hand increasing the granular size but maintaining the number of granulars for heterochromatin could also be possible in an early stage of apoptosis.
So one could only speculate without precise measurements. However, radiation and DNA repair experiments with these two cell lines and different doses and quality of irradiation did not show considerable apoptosis effects, especially if we compared the outcome with radiation experiments done on Jurkat cells which are known to be radiation sensitive and therefore show a strong apoptotic reaction.
Nevertheless without any detailed speculation we mentioned the apoptosis effects in the discussion (chapter 3.6).
6 ‘cohesion’ should be changed into ‘cohesin’ in many places.
We regret that this error caused by automatic correction in word was not completely deleted. We hope that it is corrected completely now.

Reviewer 2 Report
In this manuscript Fisher et al. demonstrated the self-organizing nature of chromatin under different conditions by using a quantifiable Imaging tool.
I appreciated the novelty of using this kind of approach, overall, the paper is well written, and I haven`t detect any language or form issues.
However, I think that the main value of this work is the novelty of the approach more than the individual results of comparing the different cell lines or conditions. I would suggest proposing the article as a brief report or a "cutting-edge" communication, by adapting the shape accordingly.
I would eliminate the "mouse model" from the comparison, given the likely rough differences between "chromatin-whole-system" in different species, pointed out by the authors themselves.
I would rather check the effect of azacitidine in human samples.
Author Response
ad Reviewer 2
In this manuscript Fisher et al. demonstrated the self-organizing nature of chromatin under different conditions by using a quantifiable Imaging tool.
I appreciated the novelty of using this kind of approach, overall, the paper is well written, and I haven`t detect any language or form issues.
However, I think that the main value of this work is the novelty of the approach more than the individual results of comparing the different cell lines or conditions. I would suggest proposing the article as a brief report or a "cutting-edge" communication, by adapting the shape accordingly.
We thank the reviewer for his very positive opinion about this manuscript and its novelty. Concerning the results of the different cell lines, we would like to mention that these detailed results are parts of a larger research project in which mainly super-resolution localization microscopy is applied. Since we plan to publish localization microscopy results for the cell types used, which will go into the detailed topology of the labelling, we think that the results shown here would be a strong support of our future publications.
I would eliminate the "mouse model" from the comparison, given the likely rough differences between "chromatin-whole-system" in different species, pointed out by the authors themselves.
I would rather check the effect of azacitidine in human samples.
Experiments with azacytidine and other chemo-therapeuticals are planned for the near future but the time given for revision is limited. So in order to show that chemical treatment could lead to similar effects like radiation treatment we included the mouse results. This is now explained in the discussion (chapter 3.6).

Reviewer 3 Report
The manuscript "Microscopic Analysis of Heterochromatin, Euchromatin and Cohesin in Cancer Cell Models and under anti-Cancer Treatment", written by Fisher EF, Pilarczyk G and Hausmann M. describes the organization/arrangement of heterochromatin, euchromatin and cohesin in different cell types (human, mouse, normal, cancer), after different types of treatments affecting chromatin (demethylation of DNA by azacytidin, gamma irradiation) and at different time points after treatment. Nuclei were analyzed by confocal microscopy after immunostaining with antibodies characteristic for euchromatin, heterochromatin and cohesin.
The manuscript is well written, with the topic presented in the Introduction, result presentation and discussion. Also, Materials and Methods are described in details, together with rationale for the type of the experiment. Discussion is very long, comparing different combinations of the cell types /treatments. It would be easier to understand all the changes if a small table could be added or graphical (like arrows) presentation of the results, or the summary (conclusions contained more general facts). Although some possible mechanisms are mentioned which could be the cause of the changes detected, possibly some additional processes could be commented, like PARP1 activity in the context of DNA damage, early apoptosis, senescence, ploidity of the cell lines, radioresistance, damage foci, etc.
line 683: rationale
Author Response
ad Reviewer 3
The manuscript "Microscopic Analysis of Heterochromatin, Euchromatin and Cohesin in Cancer Cell Models and under anti-Cancer Treatment", written by Fisher EF, Pilarczyk G and Hausmann M. describes the organization/arrangement of heterochromatin, euchromatin and cohesin in different cell types (human, mouse, normal, cancer), after different types of treatments affecting chromatin (demethylation of DNA by azacytidin, gamma irradiation) and at different time points after treatment. Nuclei were analyzed by confocal microscopy after immunostaining with antibodies characteristic for euchromatin, heterochromatin and cohesin.
The manuscript is well written, with the topic presented in the Introduction, result presentation and discussion. Also, Materials and Methods are described in details, together with rationale for the type of the experiment.
We thank the reviewer for his work and his positive opinion in general to the elaboration of the manuscript.
Discussion is very long, comparing different combinations of the cell types /treatments. It would be easier to understand all the changes if a small table could be added or graphical (like arrows) presentation of the results, or the summary (conclusions contained more general facts).
We completely agree with the reviewer that the detailed discussion is very long although we did not consider all details as the reviewer has mentioned below. For better readability we have separated the discussion in different chapters. However we agree that among all findings the major and general chromatin reactions are difficult to extract. Therefore we modified the conclusion and integrated the dominant response effects analysed.
Although some possible mechanisms are mentioned which could be the cause of the changes detected, possibly some additional processes could be commented, like PARP1 activity in the context of DNA damage, early apoptosis, senescence, ploidity of the cell lines, radioresistance, damage foci, etc.
The effects of damage foci were described in several publications which are mentioned in the discussion now. Furthermore this will be subject of other publications under preparation. Concerning early apoptosis and radio-resistance some remarks were added to chapter 3.6. Senescence and the relevance for the development of secondary cancer after harsh treatment is subject of a starting project in our group. As the conditions used in the manuscript are not as harsh as it is envisaged in the new project, we did not consider these aspects here.

Round 2
Reviewer 2 Report
I am satisfied with the author`s response to my comments.
I don`t have any other amendments to suggest.